# Water Reuse, a Sustainable Alternative in the Context of Water Scarcity and Climate Change in the Lisbon Metropolitan Area

Sofia Cordeiro [1,*], Francesco Ferrario [1], Hugo Xambre Pereira [2], Filipa Ferreira [3] and José Saldanha Matos [3]

1   Instituto de Ciências Sociais, University of Lisbon, Av. Prof. Aníbal Bettencourt 9, 1600-189 Lisboa, Portugal; fferrario@edu.ulisboa.pt
2   Águas do Tejo Atlântico, S.A., Fábrica de Água de Alcântara, Avenida de Ceuta, 1300-254 Lisboa, Portugal; hugo.pereira@adp.pt
3   Civil Engineering Research and Innovation for Sustainability, Instituto Superior Técnico, Universidade de Lisboa, Av. Rovisco Pais 1, 1049-001 Lisboa, Portugal; filipamferreira@tecnico.ulisboa.pt (F.F.); jose.saldanha.matos@tecnico.ulisboa.pt (J.S.M.)
*   Correspondence: mscordeiro@edu.ulisboa.pt

**Abstract:** Water scarcity is a driver for society to rethink water management and change the paradigm of use to a fit-for-purpose approach—i.e., separating potable water for human consumption (drinking, cooking or personal hygiene) from all non-potable uses that do not require the same quality level. In this context, urban water reuse is a relevant tool for municipalities and metropolitan areas when dealing with pressure on water resources, among several alternative water sources that can be considered in a site-specific and integrated manner. Through the available literature and specific case studies in the Lisbon Metropolitan Area, this paper explores the benefits and barriers of water reuse and intends to support local authorities in including water reuse in their water management strategies. To the best of our knowledge, this is the first paper focusing on Portugal and the Lisbon Metropolitan Area that globally examines governance, economic, legislative and social aspects regarding water reuse and presents specific implementation examples covering potable and non-potable as well as direct and indirect reuse.

**Keywords:** water reuse; climate change; urban water management; Lisbon Metropolitan Area

## 1. Introduction

Water is essential for all life forms and is a limited natural resource. Seventy-one percent of the Earth is covered by water, with saline ocean water accounting for around 97% of the total water availability. According to the AR6 IPCC report [1], terrestrial freshwater represents less than 2% of all water on Earth, a revised assessment from previous estimations of 2.5% [2]. Less than 4% of freshwater is easily accessible and available for essential ecosystem functioning and human society's water resource needs. This residual fraction of freshwater represents a total volume of about 835,000 km$^3$, mostly contained in groundwater reserves (630,000 km$^3$), with the remaining 205,000 km$^3$ being stored in lakes, rivers, wetlands and soils.

The effects of the changing climate on the water cycle are not linear but depend on several interlinked feedback effects [1]. Temperature, evapotranspiration and precipitation patterns are affected by increased GHG concentration in the atmosphere and anthropogenic climate change and, in turn, affect water availability for human activities.

Worldwide population growth and improvements in people's quality of life have led to a growth in the demand for water for human activities [3].

Agriculture represents 70% of water demand worldwide, industry represents 20% (75% of which is for energy production and 25% for manufacturing) and domestic use represents 10%. Nevertheless, there are stark regional contrasts. For example, in Portugal, agriculture represents 75% of the total water consumption despite only 15.9% of the UAA

(utilised agricultural area) being irrigated agriculture. Other Mediterranean countries, such as Spain or Greece, have a similar order of magnitude in terms of the share of total water consumption (80%) as they rely on irrigated crops. In contrast, Northern European countries can rely on rainfed farming regimes, requiring less water abstraction [4].

Global water demand has increased by 600% in the last 100 years, and it is expected to increase by 20% to 30% by 2050, reaching 5500 to 6000 km$^3$ for all uses (industrial, domestic and agricultural). Such growth will be unequally distributed in terms of sector and geography, leading inevitably to water stress and injustice [3]. Urban water consumption in particular has increased constantly over the last few decades [5].

Besides the increase in global water consumption, water availability is also shrinking because of quality deterioration and spatial and temporal distribution changes. The predicted and observed scarcity of surface water has led to an increase in demand from groundwater systems, which are globally already near the maximum sustainable levels [6] and rapidly deteriorating. This phenomenon is caused by over-extraction and saline intrusion, occurring mainly in coastal areas, where the increase in water demand is driven by concentrated population growth [7].

Anthropogenic activities are responsible for increasing demand, shrinking freshwater availability (via climate change) and water contamination [8]. Pollution is deteriorating the quality of available water resources and is expected to increase precisely due to socio-economic stressors. In fact, pollution is mainly caused by a lack of sanitation, nutrient loading (predominantly phosphorus and nitrogen from agriculture, which are already surpassing sustainable levels (as per [9]), chemicals (linked to intensive agricultural processes and the use of herbicides, insecticides and fungicides) and novel and emergent pollutants, such as pharmaceuticals, hormones and personal care products [8]. Additionally, ecosystem changes driven by land-use changes (e.g., the disappearance of wetlands and forests) are expected to have dramatic effects on the hydrological cycle and, consequentially, on water availability [10].

Several alternatives to the use of the available superficial water have been assessed and adopted worldwide, from rainwater collection to the exploration of new groundwater resources. Regarding rainwater collection, rain regimens are unpredictable. They are also intensely affected by climate change, with climate hot spots such as the Mediterranean expected to suffer a reduction in precipitation by up to 40% [1]. Groundwater reserves, when used intensively without allowing for adequate recharge, rapidly become unsustainable, even if abundant. In fact, more than 30% of groundwater systems are already in distress [11] because of rapid consumption without actual knowledge of overall water availability [12].

In this context, considering water scarcity and climate change, water reclamation and reuse should be seen as a sustainable alternative that avoids further abstraction and is independent of climate uncertainty, particularly at the urban level. Cities tend to exert significant pressure on water resources due to the concentration of their populations, industries, and productive activities. Hence, water reuse is a relevant adaptation tool for municipalities that aim to increase their long-term resilience. Additionally, because of the concentration, economies of agglomeration can be reached in urban water reuse.

This paper explores the benefits and barriers of urban water reuse with the goal of supporting municipalities to include this sustainable approach in their water strategy. Our conceptual analysis and presentation are supported by evidence from some cases of water reuse in the Metropolitan Area of Lisbon, Portugal.

There have been previous works on water quality standards or regional flows in the region, but the broader perspective we take in this article, encompassing governance, economic, legislative and social aspects and focusing on the situation in Portugal and the Lisbon Metropolitan Area, has not yet been systematically approached. As the effects of climate change and the consequent water scarcity in Mediterranean climates become harsher, there is a political will to create conditions that foster the use of alternative water sources. We aim to contribute to the discussion of the steps to be taken in order to make

water reuse projects feasible for more sustainable and integrated water management in the region.

## 2. Water Reclamation and Reuse as a Sustainable Alternative

According to the principles of sustainable development, a circular economy approach can and should play a key role in water management. In fact, the reuse of treated wastewater, catering to specific uses, would avoid further abstraction from strategic water reserves.

Water reclamation and reuse have the potential to (1) reduce freshwater and groundwater consumption for both potable and non-potable uses; (2) increase the capacity to meet water demand over time; (3) reduce water demand pressure on aquatic ecosystems and preserve water quality in waterways; (4) reduce investments in water control structures; and (5) improve water management sustainability [13].

Water reuse applications can be divided into the following seven categories [14] by order of relevance:

– **agricultural irrigation**—the largest current use of reuse water throughout the world, with the most significant impact potential in both industrialized countries and developing countries;
– **landscape irrigation**—irrigation of parks; playgrounds; golf courses; freeway medians; landscaped areas around commercial, office and industrial developments; and landscaped areas around residences;
– **industrial use**—primarily for cooling and process needs, with a high impact potential since cooling water creates the single largest industrial demand;
– **groundwater recharge**—groundwater replenishment by the assimilation and storage of reuse water in groundwater aquifers or the establishment of hydraulic barriers against salt-water intrusion in coastal areas;
– **recreational and environmental uses**—non-potable uses related to land-based water features, such as the development of recreational lakes, marsh enhancement and streamflow augmentation;
– **non-potable urban uses**—including fire protection, air conditioning, toilet flushing, construction water, and the flushing of sanitary sewers, typically in proximity of wastewater reclamation plants and/or coupled with other ongoing reuse applications such as landscape irrigation;
– **potable reuse**—although the technology exists, only a few projects have been implemented worldwide (e.g., the classical example of Windhoek, Namibia [15]).

Of these, potable reuse requires the highest level of quality through a higher level of treatment, which can be achieved in two different ways. (1) Direct potable reuse consists of the reintroduction of treated wastewater directly into distribution; and (2) indirect potable reuse happens through an environmental buffer, as treated wastewater is introduced into the surface water or groundwater and mixed with the waterbody, becoming available for abstraction again [13].

Urban water reuse should become an alternative water source to meet urban demand, at least for non-potable uses such as the irrigation of green spaces, street and vehicle washing, firefighting, toilet flushing and climatization. The validity of this option is justified by the high urban water consumption and by the fact that domestic wastewater treatment plants tend to be located in the vicinity of city centres. Moreover, when used for urban landscaping, reuse water has the additional benefit of containing nutrients, such as phosphorus and nitrogen, thus requiring a less intense use of fertilizers, contributing to another level of circularity and being an important advantage of the approach. In turn, the reduced use of fertilizers positively reduces pollution and increases the quality of waterways and connected biospheres [13]. Reuse water can also be used for recreational and environmental uses, such as artificial lakes and marsh enrichments [16], positively affecting urban resilience to climate change.

Depending on the use, different levels of treatment are necessary [13]. As non-potable reclamation and reuse systems are typically designed for peak flows in the driest periods,

both in terms of quantity and quality, there is an extra capacity in the wet season that could be used to recharge aquifers, thus helping to reduce groundwater distress level [11,12].

It should be noted that the treatment of wastewater to a level of quality that allows for its reintroduction into the system reduces pollution and the negative environmental impacts linked to wastewater [17].

The first element of complexity arises in understanding that reuse water should be a substitute source, rather than a way to support increasing water demand. The rationale is that cities will become more sustainable by increasing water reuse, not necessarily increasing the overall water consumption. The availability of an additional source might create the perception of abundance and increase consumption. The presented dynamic is a well-known effect in the energy sector, known as the rebound effect: improvements in energy efficiency and renewable energy availability can lead to higher, rather than lower, overall energy consumption. As a general conclusion, the focus of public policy should always be, first and foremost, on water savings.

## 3. Gaps and Drivers for the Success of Water Reuse Projects

Currently, we consider that three main factors contribute to the low adoption of reclamation and reuse as a viable alternative, particularly in Europe: (1) the ease with which new water intakes are licensed, in many cases regardless of the status of the reserves and without control over consumption; (2) the reduced capacity to monitor and apply fines for illegal intakes by environmental authorities; and (3) the low costs and tariffs, not only on water itself but also on the concessions for intakes and on the resource use.

Regarding licensing and control over consumption, as an example, in Portugal, according to public data from the Portuguese Environment Agency [18], only in the period from June to November 2017, a year marked by an extreme drought, 8194 new groundwater intakes and 409 freshwater surface extractions were licensed or legalized (Figure 1). These numbers have been growing, with increasing degradation of water bodies and greater energy consumption. Empirically, it is easy to understand that the cost of water for agricultural and urban irrigation and other uses is directly proportional to the investments that citizens and companies are willing to make in water use efficiency or in alternative sources, such as water reuse or water transfers. Clearly, low costs or tariffs that do not reflect the cost of water scarcity will always be a barrier to implementing water reuse systems.

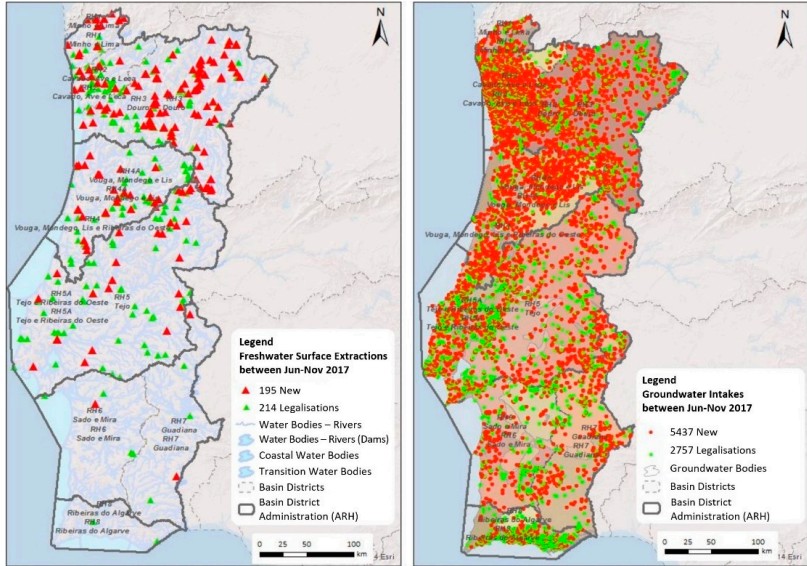

**Figure 1.** Geographical distribution of surface and underground catchments, with a Water Resources Use Permit (TURH, Título de Utilização de Recursos Hídricos) issued between 1 June and 30 November 2017, distinguishing new catchments (in red) from regularization situations (in green). Source: Portuguese Environment Agency [18].

### 3.1. Governance and Institutional Collaboration

Water reuse projects require collaboration and a joint assessment of needs and impacts between the different stakeholders in the water sector. Governance systems can be key to either overcoming barriers or becoming barriers themselves. Good governance is a critical tool in integrated water resource management (IWRM) frameworks as it facilitates successful policy, regulatory and legislative initiatives, financing and public acceptance, to mention a few relevant issues. Ref. [19] identified four types of ownership structures in reuse water management from an in-depth survey of a large number of European water reuse projects:

- A single entity (water and sewerage under single responsibility);
- A water company managing reclamation;
- A wastewater company managing reclamation;
- An ad hoc project structure developed coherently with local circumstances.

Based on the literature, survey results and an international workshop, the authors of [19] reported that a clearer institutional arrangement was needed for the increased adoption of water reuse. Although stressing that project ownership is not the real issue via an analysis of successful case studies (but is more tied to liability, access to finance and cost allocation), the study suggests that a tight compartmentalisation of water supply and sanitation services has the potential to result in poor institutional arrangements for global, integrated water cycle management in general and water reuse in particular. This leads to time lags between feasibility studies and the actual project implementation. Governance is also key to overcoming public acceptance issues.

In the Lisbon Metropolitan Area, the different phases of the urban water cycle, from the water supply to sanitation services, are performed by a total of 19 distinct entities, from municipalities, directly or through municipalised services, to municipal or intermunicipal companies, or under a concession (to a public or private entity). With such a complex governance architecture, in order to take advantage of potential metropolitan synergies, strong overarching strategies and policies are required.

### 3.2. Economic Evaluation

In general terms, the value of water is rarely captured in consumer prices. Many elements contribute to the final overall value of the resource: its primary use, the state of the network, the availability of the resource, and local environmental and socio-economic conditions [20]. The consequence is that users pay a tariff that rarely allows for cost recovery, limiting the level of investment in the infrastructures [21].

Reaching full cost recovery is even more complex when dealing with reuse water systems due to costs linked to water treatment and dedicated distribution infrastructure (pipes, boosters and water tanks). An additional component to this imbalance is related to the fact that environmental costs are not included in the costs of freshwater extraction and consumption [22]; hence, methodologies that consider external costs and impacts (environmental and social, for example), and not only the internal costs and impact, will be critical for a correct assessment of the feasibility of water reuse projects [23].

In many if not most cases, when using a classical, narrow cost/benefit analysis, water reuse is considered more expensive than abstracting, treating and distributing surface water or groundwater. In these cases, keeping water reuse prices lower than those of conventional water sources could imply that cost recovery is not met by the water reuse scheme alone, potentially leading to governance and policy issues given that the ownership and management of water reuse and freshwater catchment and distribution might be dispersed among different entities. Therefore, the business models have to be integrated through policy.

A comprehensive cost recovery approach should be taken, considering the following: (1) the tariffs for water reuse (that can be applied on a volumetric basis to "direct" users of reuse water); (2) the tariffs for wastewater collection and treatment (which already cover part of the existing cost structure to treat the effluent); and (3) the tariffs for freshwater

supply. The latter could cross-subsidize water reuse, considering that water reuse will protect drinking water reserves and thus increase water security. Many countries have in place legislation that supports these subsidies (especially at the European level) [21], such as those applied in the Prato industrial area [24].

To foster water reuse, its pricing should encourage it, when compared to that of freshwater. In extreme situations, such as the classic water reuse example of the city of Windhoek, Namibia, the cost of reuse water for direct potable reuse is lower than the cost of surface water because of absolute scarcity in the region, but also due to the distance to viable sources and particular geophysical characteristics [21,25].

As a possible solution, we argue that water tariffs could be built/revised to include a component directly allocated for water reuse after the initial use and adequate treatment of the resulting wastewater. Such change would incentivise adopting reuse water (which would be cheaper) rather than abstracting more water from surface or groundwater reserves, as new abstraction would always pay for the reuse component in the tariff. This change in regular water tariffs should be complemented by creating a simplified tariff system for water reuse, taking into account varying quality needs in a fit-for-purpose approach. Including the water reuse costs in tariffs is not the only solution; end users should also incur some of the costs, depending on project specificities. One good example is the municipality of Prato, in Italy [24], which is surrounded by a textile industrial district with high water demand but is further away from the WWTP catering to the city centre's wastewater treatment needs. An association of the companies in the industrial district financed the infrastructure to convey treated urban wastewater to the industrial district, as they were the main users, providing a business model tailored to the area's specificities.

### 3.3. Water Reuse in the European Union Policy Framework

The need to better manage water resources and adapt to increasing water scarcity underlies all environmental EU policy instruments, which generally aim at sustainable resource management. In European law, water reuse is identified and fostered as a solution in two European Union directives. The focus is mainly on agricultural irrigation, and the directives do not establish quality standards. The Urban Waste Water Treatment Directive—UWWD, Council Directive 91/271/EEC concerning urban wastewater treatment [26]—enunciates the principle of reuse in Article 12: "Treated wastewater shall be reused whenever appropriate". Although it lacks clarity in the definition, maybe purposefully to allow Member States to create or adapt the most efficient frameworks for their specific situations, it creates a favourable context by considering it desirable and acceptable to reuse treated wastewater. The later Water Framework Directive—WFD, Directive 2000/60/EC of the European Parliament and of the Council establishing a framework for community action in the field of water policy [27]—defines objectives for the quantity and quality of water and the instruments to promote sustainable water use. Wastewater is included as one of the potentially most impactful supplementary measures in this programme (under Article 11, Paragraph 4 and Annex IV, Part B), as well as aquifer recharge (which can be done with reuse water).

The EU prioritizes savings and efficiency as sustainable water management measures, above investing in alternative water sources and additional water supply infrastructures. The 2007 Commission Communication on Water Scarcity and Droughts [28] proposed a water hierarchy whereby additional water supply options (e.g., desalination) should only be considered after all other improvements in efficiency on the demand side are exhausted, including water reuse as an alternative source. However, it was only used to a limited extent in the EU, notably due to a lack of environmental/health standards for reuse and the potential obstacles to the free movement of agricultural products irrigated with reused water.

The regulation on minimum quality requirements for reuse was approved in 2020 [29] and is a critical development, but targets specifically agricultural irrigation to address safety issues affecting free trade in the EU. It notably missed an opportunity to create the

same standards for other possible uses that can benefit from water reuse as an efficient water management measure, such as urban uses. If the free circulation of agricultural products was the driving force for the definition of these common standards, the free circulation of people and the concern for their common safety standards in public spaces could, and arguably should, have also been considered. This could have been accomplished by equating urban reuse in public spaces without restriction of access (gardens or parks) to the most stringent minimum standards defined for agricultural use—water quality class A, required for "All food crops consumed raw where the edible part is in direct contact with reuse water and root crops consumed raw", as defined in Annex I of the regulation [29].

The proposed revision of the Urban Waste Water Treatment Directive (UWWTD) [30] reflects present concerns related to the lack of economic sustainability of services and the environmental degradation of receiving waters, aiming at energy neutrality at the national level for wastewater facilities above 10,000 p.e. Another goal, the reduction in pollution due to rain waters in large agglomerations, will require the implementation of integrated urban water management plans (the use of source control approaches or constructed wetlands, among other nature-based solutions, is seen as an attractive method for overflow treatment in Mediterranean urban areas) [31]. Furthermore, the UWWTD revision proposal envisions an overall reduction in water pollution, imposing more stringent limit values to treat nitrogen and phosphorus. Limit values are also established for micro-pollutant control depending on the served population and, in certain cases, the receiving waters. A system that increases the producer's responsibility for targeting pharmaceutical and personal care products (the two main sources of micro-pollutants) will be set to cover the associated additional treatment costs.

As is evident from the evolution of the UWWTD, urban wastewater will continue to be subject to more demanding and efficient treatment schemes, increasing the availability of higher-quality treated wastewater. This may allow for relevant improvements in the feasibility assessment of water reuse projects, namely from an economic perspective. If these technological developments are followed by a streamlining of the processes leading to water reuse, notably licensing, the adoption of water reuse could be significantly increased. Simplified licensing procedures would not only be adequately reflecting the lower risks from more demanding wastewater treatment levels, but also facilitating reuse as an important adaptation measure. Promoting water reuse is also an effective way of adapting to the more intense drought regime which is increasingly prevalent in Mediterranean climates due to increasing climate change effects.

### 3.4. National Legislative and Policy Instruments

In the EU context, which provides a framework but leaves ample room for Member States to make decisions, national regulations and legislation play a key role as instruments to foster or hamper the implementation of water reuse projects. Pereira [32] recently reviewed the legislative framework evolution for wastewater reuse in Portugal, providing guidelines for specific legislative needs.

A relevant European example is the Spanish Royal Decree 1620/2007 [33] establishing the legal regime for reusing wastewater. It defines the conditions and standards for wastewater reuse with a fit-for-purpose approach, as well as an economic model (Chapter III) and all the permit and concession processing (Chapters IV and V). The Ministry for Environment, Rural and Sea Areas (Ministerio del Medio Ambiente, Medio Rural y Marino) also published a guide supporting the application of the decree [34]. With a complete approach encompassing environmental and economic aspects, this piece of legislation was an important driver for success and explains why Spain is one of the European countries with the highest volume of wastewater reuse [19,35].

In Portugal, the guiding policy principles for water and water resources management reflect the European Directives. When transposing the Urban Waste Water Directive [26] to Decree-Law 152/97 [36], the legislator followed the directive in saying that management utilities "should assess the production and distribution of treated wastewater apt for

reuse, as an alternative to the rejection in the environment, whenever the solution is technically, economically and environmentally feasible", but failed to define the "technically, economically and environmentally feasible" conditions, which does not encourage reuse. In the regime for the use of water resources, Decree-Law 226-A/2007 [37], reuse is incentivised when mentioning it as a complement to water abstraction for the irrigation of public gardens or golf courses (Article 44, Paragraph 3). The same article states that catchment for irrigation in large areas (more than 50 ha) has to comply with the efficiency rates established in the National Program for the Efficient Use of Water, PNUEA [38]. This program generally incentivises reuse, albeit at the time of its publication, specific guidelines, regulations and legislation were still lacking. At the same time, the programme is pessimistic regarding reuse project implementation, mentioning that it "requires high investments (tertiary treatment, disinfection, duplication of distribution networks) that will hardly pass the test of a cost-benefit analysis, not to mention that adequate uses are limited to washing and irrigation in the vicinity of WWTPs (e.g., golf courses, orchards)". The mentioned cost/benefit analysis does not account for the full environmental value of water, nor the economic costs of scarcity and water stress.

Specific legislation on quality was only recently published in Decree-Law 119/2019 [39] which establishes quality standards on a fit-for-purpose basis and defines the terms for licensing production and use. It builds on the European Regulation and the preexisting national norms but notably lacks an economic model and instruments that could promote project implementation. As it currently stands, this legislation is a much-needed instrument regarding health and environmental safety, although it still leaves ample room for interpretation in its application. However, it will not be a driver by itself to increase water reuse, mainly because it lacks the complementary economic and business model guidelines which would create a more secure framework for investments.

*3.5. Social Acceptance of Different Types of Water Reuse*

When properly treated, reuse water poses negligible health risks [40]. However, public acceptance is not without controversy, with aversion feelings amplified when the intended use involves human contact [41]. Studies conducted in North American cities [42,43] found that consumers were more likely to prefer not knowing or not having to choose, especially for the more environmentally conscious. When given a choice to bid for wine made from grapes irrigated with reuse water, conventional water, or an unspecified water type, participants in a lab-in-the-field experiment by [42] showed a higher willingness to pay for the unspecified type, an effect more pronounced when respondents were presented with a message explaining the positive impacts of this choice on drought issues. In a survey of more than 600 participants in three North American cities, [43] found that there is a clear variation in preferences between water used for lawn irrigation, food irrigation and drinking and that the reluctance to consume reuse water is particularly evident. In contrast, the likelihood of preferring reuse water for lawn irrigation is the highest. Interestingly, the study found that participants in more water-abundant regions are not less likely to express a preference for reuse water.

In a recent study in Lisbon, Portugal, however, the results were more encouraging [44]. The study, in the scope of the CEMOWAS2 project (http://cemowas2.com/en/ (accessed on 25 July 2023)), comprised focus groups, surveys and interviews, and aimed at assessing awareness of water scarcity, the value of water and water reuse, as well as receptiveness to various uses, potable and non-potable. The concept of water reuse is highly accepted and recognised as "natural" as a management tool, with the highest acceptance for non-potable outdoor uses, such as irrigation or street and vehicle washing. Participants also showed high confidence in consuming horticultural produce as long as legislation is in place for their safety. For human consumption, interestingly, although 50.1% of the interviewed Lisbon residents in the study were undecided (expressing that they were not sure if potable reuse will be possible in 20 years), 40.8% felt comfortable that it would happen, with as many as 9.9% expressing that they thought it was already in practice in Lisbon. Only 9%

of the interviewed expressed disbelief in potable water reuse (3.6% expressing they felt it would never happen and 5.4% that it would not be financially viable).

## 4. Water Reuse Cases in Lisbon Metropolitan Area

As previously mentioned, water reclamation is a crucial tool for urban water management. At the European level, today, on average, water stress affects about 20% of the territory and 30% of the total population [45]. In this context, about 200 municipalities have implemented water reuse projects, also due to the support of the European Commission [19]. In this section, four cases from the Lisbon Metropolitan Area will be presented: (Section 4.1) agricultural indirect reuse in Loures; (Section 4.2) trade and services' direct non-potable reuse for HVAC in a large retail store in Loures; (Section 4.3) Lisbon as a case study in evolution, addressing many of the complexities of water reclamation and reuse in urban settings; and (Section 4.4) a proof-of-concept for indirect potable reuse for beer production.

### 4.1. Agriculture: Indirect Non-Potable Water Reuse in Várzea de Loures

In recent years, some pilot projects of direct water reuse in agriculture in Portugal have been developed, namely in an orchard of pomegranate trees and vineyards in the Alentejo and Lisbon (for example, the AQUA-VINI Sustainable and SUWANU EUROPE projects).

However, reuse for agricultural irrigation also happens indirectly, with water abstraction for irrigation downstream of the discharge point of treated effluents from WWTPs. This type of indirect reuse represents the largest use per volume, but it is difficult to measure accurately, and therefore there are no official data.

This is the case of the floodplain of Loures (Várzea de Loures), in the Lisbon Metropolitan Area. Loures is a municipality of 201,590 people located north of Lisbon, in the Tagus river basin, where the Frielas WWTP is located. This WWTP serves as the indirect source of reuse water for irrigation in the floodplain. The agricultural water catchment is in the Trancão river, downstream of the WWTP discharge point, without any additional treatment beyond what is necessary to comply with the discharge license ($CBO_5$ <= 25 mg/L, SST <= 35 mg/L, *E. coli* <= 2000 u.f.c/100 mL).

The Hydro-Agricultural Exploitation of Várzea de Loures has not undergone many changes in its cultures over the last twenty years, and the main crops are tomatoes and horticultural crops (mixed vegetable gardens (55 ha), corn (60 ha), tomato (246 ha) and onion (5 ha)). The remaining area is cultivated with autumn/winter crops (cereals, hay and pastures) that are not irrigated.

The farmers of the Floodplain of Loures have historically collected water from streams in the floodplain, although using other technologies. Typically, small weirs installed in the streams divert the water to the existing drainage ditches from which interested farmers collect the water to irrigate crops. With the construction of the first WWTP of Frielas, inaugurated in 1967, water became available more regularly, including in the dryer semester, from April to September. According to Águas do Tejo Atlântico [46], the volumes discharged, with potential indirect reuse, are close to 21 million $m^3$ per year (20.99 million $m^3$ in 2021 and 20.90 $m^3$ in 2022). However, these values are affected by seasonality, with higher discharges in the months with the highest precipitation, from October to March. It should be noted that the discharge flow is crucial to maintain the necessary watercourses in dry years and that the discharge volume is lower in the hot and dry months when water for irrigation is mostly required (Figure 2).

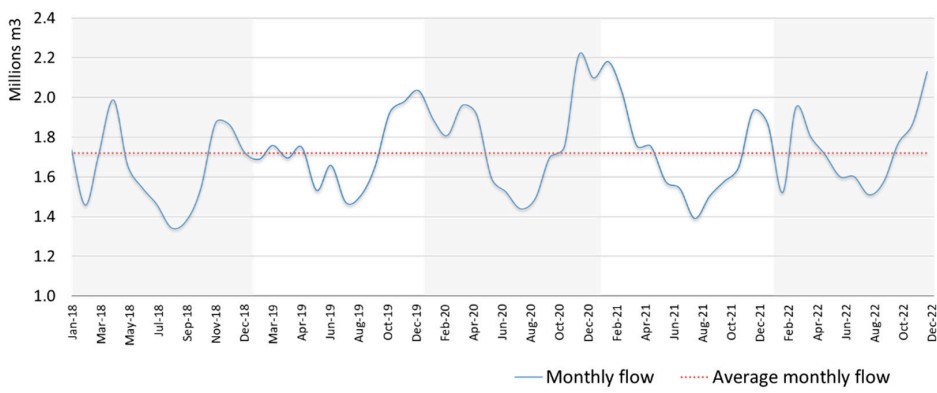

**Figure 2.** Monthly distribution of discharged treated effluents from Frielas WWTP over the course of 5 years, between January 2019 and December 2022. Average monthly flow of the 5-year period is represented by the dashed red line. Data from Águas do Tejo Atlântico [46].

Considering that the floodplain of Loures does not have an irrigation system installed, it is impossible to accurately quantify the volumes of reused water. The volumes estimated by the Association of Beneficiaries of Loures in 2021 were 2.9 million m$^3$, representing only about 14% of the volume discharged by the WWTP.

The indirect reuse of water at this site is not subject to any quality monitoring and has been common practice for more than 50 years without any record of public health problems.

*4.2. Trade and Services: Direct Non-Potable Reuse for HVAC at the IKEA Store in Loures*

Since 2019, the IKEA store in Loures has been reusing treated wastewater from the Frielas WWTP for its HVAC system. The proximity of this large retail store to the WWTP, and the fact that the project was designed from scratch to use this alternative water source, were crucial aspects for the viability of the reuse project (Figure 3).

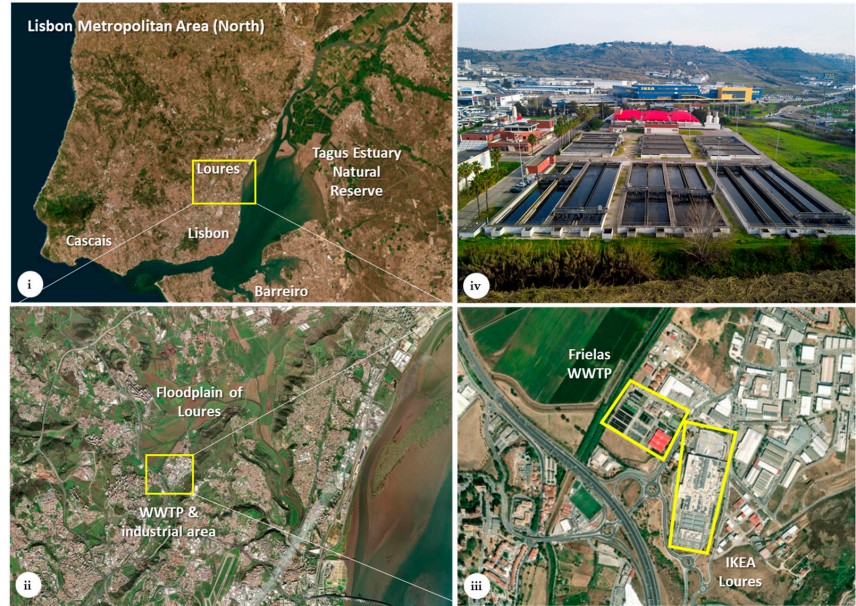

**Figure 3.** Frielas wastewater treatment plant in Loures, Lisbon Metropolitan Area and the IKEA store. (**i**) Location of Loures in the Lisbon Metropolitan Area; (**ii**) Loures area, with the industrial area (highlighted in the yellow square) where the WWTP and IKEA are located and the Floodplain of Loures (see Section 4.1); (**iii**) Industrial area of Loures, with the Frielas WWTP and IKEA Loures, located in the immediate vicinity; (**iv**) Frielas WWTP tanks (in the front) and the IKEA Loures store (in the back). The HVAC system at IKEA Loures exclusively uses reused water from the WWTP of Frielas, with an estimated 360,000 m$^3$ of annual consumption, representing 1.7% of the WWTP capacity.

The reused volume is approximately 360,000 m$^3$ per year—corresponding to about 1.7% of the total wastewater treated in the Frielas WWTP. The water supplied is approximately at 22 °C, and its quality is class B. This quality level, defined in European and national legislation, is perfectly suitable for this purpose, as the water in the HVAC system is permanently in a closed circuit, without direct contact with the public or workers, nor indirectly with any equipment with which the public or workers may have contact.

*4.3. Urban Uses: Direct Non-Potable Reuse in Lisbon and Potential for the Metropolitan Area*

The case of the Lisbon Metropolitan Area (LMA) is interesting as currently, it is not under intense water stress, although there is a 75% dependency of the region on its main reservoir (Castelo de Bode, located more than 150 km away from Lisbon), and the southern part of the metropolitan area depends on groundwater sources, which already present signs of overexploitation. However, the country is expected to suffer from water scarcity due to anthropogenic climate change [47].

The Lisbon Metropolitan Area represents only 3.3% of the national territory but hosts about 25% of the national population, 30% of the companies and about 33% of the jobs [48]. These figures can give an idea of the pressure on water resources in the metropolitan area. So, although technically not under extreme water stress, increasing reuse in the region could allow for the better use of the reserves which currently supply the Lisbon Metropolitan Area. Increasing water reuse would contribute to strategically preserving these reserves in order to endure more intense, prolonged and frequent drought periods that may be expected in the future.

Since 2004, the Lisbon Municipality and utilities have jointly developed a water management strategy to address its water pressure, based on the following: (1) reducing systemic water losses; (2) actively managing demand; and (3) reusing treated wastewater for non-potable uses [48].

In a detailed analysis [49] of the water flows in Lisbon Municipality (the largest of the 18 municipalities in the Lisbon Metropolitan Area), non-potable urban uses represented 75% of total potable water consumption (54% for green space irrigation and 21% for street washing), and the water consumed by the Municipality represented 15% of the total consumption in the city. With efficiency and leakage control measures, Lisbon was able to reduce its consumption by 50%, a reduction from 8.2 to 4.2 million m$^3$ per year, between 2014 and 2018 [50]. Urban uses now represent 7% of the overall consumption of potable water in the city [50], indicating an overall increase in city demand, most likely due to economic activity. The non-potable urban uses still represent the majority of demand (75% of municipal consumption, 3 million m$^3$), so there is a high potential for water reuse to replace the potable freshwater currently used for those purposes.

As so, the Lisbon Municipality devised a strategic plan for water reuse at a city scale [51,52] (Figure 4) involving the construction of a distribution network for reuse water from the city's three WWTPs (Alcântara, Chelas and Beirolas) to nearby major points of non-potable urban consumption.

Different uses have been identified depending on the proximity to each of the three WWTPs. For instance, the Alcântara WWTP is connected through a water reuse distribution system to the city centre, where the reuse water can be used to irrigate green spaces and for street cleaning (Figure 5). Another example is the Beirolas WWTP, which has recently (as of March 2022) started to provide reuse water to irrigate the adjacent Tejo Urban Park green area (up to 400,000 m$^3$/year). In this case, an investment of approximately EUR 700,000 was made to partially increase the treated water quality, adding microfiltration followed by ultrafiltration with membranes to the existing tertiary treatment process (Figure 6).

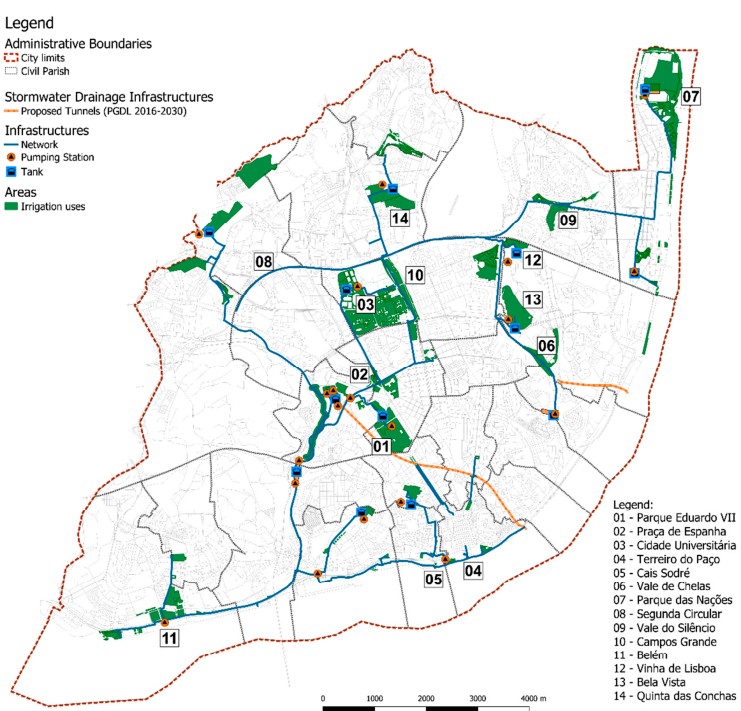

**Figure 4.** Interventions planned in the "Lisbon Strategic Plan for Water Reuse" (2020–2025) [51,52]. The plan was presented in 2019 and aimed to promote, by 2025, the reuse of 1.6 million m³/year of water, after the construction of about 55 km of main pipelines, 13 water tanks and 19 pumping stations. Reprinted with permission from Ref. [51]. 2021, Cui et al.

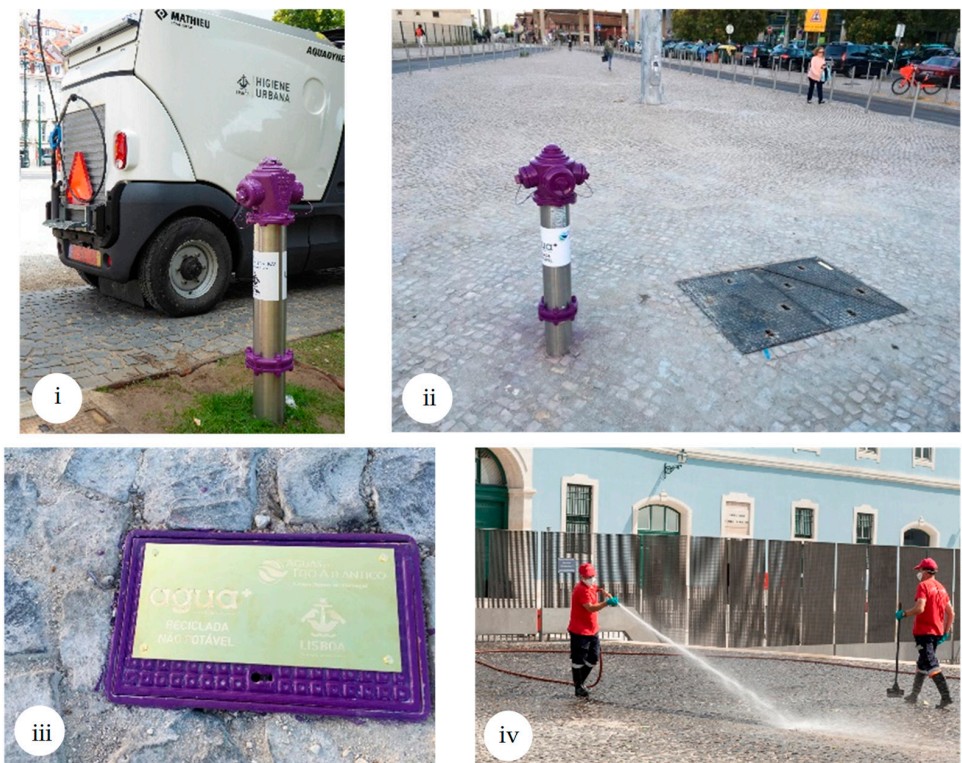

**Figure 5.** Water reuse hydrant network in downtown Lisbon. (**i,ii**) First water reuse hydrants installed in downtown Lisbon area; (**iii**) Water reuse ground hydrant in downtown Lisbon; (**iv**) Urban hygiene staff washing a street using treated wastewater in downtown Lisbon.

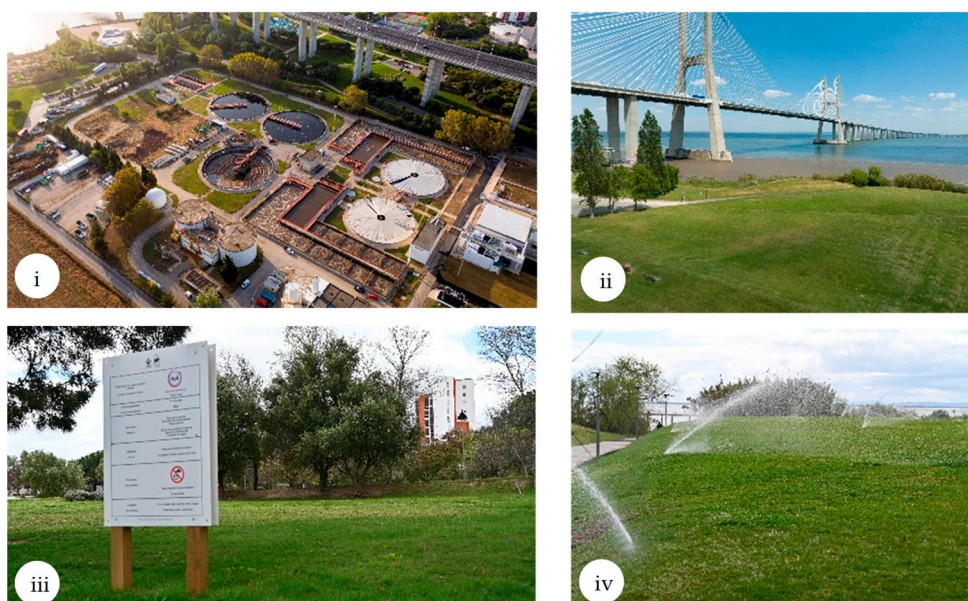

**Figure 6.** Irrigation at Tejo Urban Park, in Parque das Nações neighbourhood, the first open-access urban park irrigated with reused water in Portugal. (**i**) Beirolas Wastewater Treatment Plant; (**ii**) Tejo Urban Park, the first urban park irrigated with treated wastewater; (**iii**) First irrigation at Tejo Urban Park with reused water from Beirolas WWTP; (**iv**) Signage at Tejo Urban Park, as a safety barrier when using reused water for irrigation in open-access spaces.

Although the Lisbon water reuse strategic plan is already being implemented, Tejo Urban Park is still the only licensed irrigation project in a public space because of a lengthy permit issuing process by the National Environment Agency. The process to obtain a permit is very demanding and quite stringent, probably since this was the first-ever project of irrigation with reuse water in a public space with no restrictions of access. The irrigation of golf courses or agricultural exploitations such as citrus orchards are already well documented in Portugal [53] and supported by the recent water quality for reuse legislation [39]. However, for urban uses, safety concerns arise, given the higher potential for direct human contact with reuse water, which has led to very low use so far, and mostly in street cleaning, not in irrigation. There is a paradox in these safety concerns since the highest water quality standards for reuse [39] are more stringent than the highest quality standards for safe human access to coastal or river bathing waters [32].

### 4.4. Food and Beverage Industry: Indirect Potable Reuse for Human Consumption with VIRA Beer as a Proof of Concept

As a communication and environmental awareness project of Águas do Tejo Atlântico, VIRA beer is a craft beer produced with wastewater treated at the Beirolas WWTP (Figure 7). This treated wastewater is labelled "água+", which translates as "water+": for every 1000 L of beer, 2000 L of "água+" are required. A fine-tuning process was added to the conventional tertiary treatment process (Figure 7), including the following: (1) ozone oxidation to disinfect the effluent, destroying the cell membrane of microorganisms and degrading soluble contaminants; (2) reverse-osmosis filtration to remove salts, nutrients, metals and organic micropollutants; and (3) constant analytical monitoring to ensure a quality compatible with potable uses [54]. The project, which was not aimed at commercialisation, was awarded the Water Reuse Europe Innovation Prize in 2021.

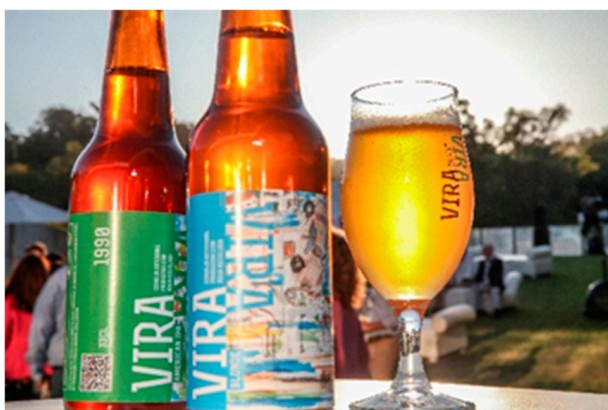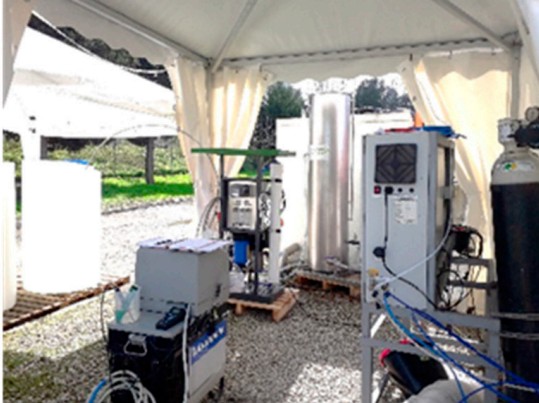

**Figure 7.** VIRA beer, produced with reuse water (**left**), and the ultrafiltration and reverse-osmosis system to produce the water quality required for VIRA beer production (**right**).

The production of this beer with reuse water following the appropriate treatment is a proof of concept that the technology for potable reuse exists and is safe under existing legislative and regulatory frameworks in the European Union, also demonstrating the economic potential for the installation of a unit (or a cluster of companies, as has begun to exist in Lisbon) of craft beer production in the vicinity of WWTPs. Similarly, cost savings are also achievable for other economic activities requiring large volumes of water for their processes.

## 5. Discussion

As the reviewed literature and case studies show, water reuse is a valuable tool to reduce urban metabolism inputs and the pressure on water resources intrinsic to population agglomerations. This discussion aims to highlight the benefits and barriers of water reuse to guide local authorities when approaching reuse strategies.

The first relevant point is that water reuse is not a stand-alone solution. It has to be part of a wider urban water strategy with multiple goals, including the reduction in water demand and close monitoring of network losses. At the same time, the whole strategy has to be contextualised in a city's dynamic and changing social and natural environment. The evidence from the case studies indicates that the choice to include water reuse in an urban water strategy often springs from a water crisis and that it is crucial to address it in an integrated way. Factors to consider are the overall water availability, both in the short and long term, the presence and location of WWTPs, the possibility of building a secondary distribution system for water reuse, the potential risks in the locations where its use is considered and the mitigation of these risks.

The construction of a reuse water distribution system, including pipes, pumping stations, storage tanks and chlorination installations, is the most complex and costly element of the strategy and has to be addressed according to the potential use and availability of wastewater to determine its feasibility. Lisbon has devised a strategy to connect the WWTP to high-consumption areas, such as urban parks (for irrigation), via a new pipe network. This network could supply other uses in the future for facilities with high water demand, such as retail outlets or industry. This type of investment is more costly than approaching the potential reuse in the immediate vicinity of WWTPs, as is the case of the Hydro-Agricultural Exploitation of Várzea de Loures (Section 4.1) or the IKEA store in Loures (Section 4.2), in which there is indirect or direct water reuse from the Frielas WWTP or the irrigation of Tejo Urban Park (Section 4.3), with direct reuse from the Beirolas WWTP, without the need for the construction of a costly and complex infrastructure.

A less expensive alternative for small volumes consists of transporting reused water by trucks or smaller vehicles, which allows for access to areas where the construction of the physical network would be too expensive or not feasible. Since using vehicles might also

have negative climate impacts, such as GHG emissions with combustion engines, this will not always be a sustainable solution, especially at larger scales. The use of electric vehicles charged with renewable energy sources could be a future improvement. Nevertheless, new water distribution infrastructures could also be implemented at lower costs by using ongoing sanitation infrastructure renovation to include reuse water distribution and using surface or underground infrastructure when developing or renovating wide avenues, highways, subway tunnels, cycle paths or pedestrian dividers well integrated into the urban design.

All options have fixed and variable costs that are not negligible. As previously indicated, these costs have to be considered on top of the cost of wastewater treatment. Hence, the overall cost of water reuse tends to be higher than that of freshwater, if the full cost of treatment is considered. If not subsidised, reuse water might not be competitive, particularly if negative environmental externalities are not considered (e.g., continued abstraction from unsustainable groundwater or surface water resources). However, since wastewater treatment up to determined quality levels for environmental discharge is already mandatory under EU Directives, only the additional costs of infrastructure to distribute and the additional treatment in a fit-for-purpose approach, if required, should be considered when assessing the feasibility of water reuse projects.

The evaluation of costs and subsidies needs to be integrated into the financial part of urban and metropolitan water strategies: part of it can be covered by internal and/or external sources, and part can be compensated by increasing the costs of traditional water resources' usage.

Finally, another element of complexity is public acceptability, willingness to pay and willingness to access the service. Wastewater treatment and reuse water may find resistance in the population. A lack of understanding of the treatment process magnifies the perceived contamination risk. In this sense, local authorities and utilities have the responsibility to monitor the quality of the reuse water closely and put in place a communication strategy that allows the population to understand the potential risks, the corresponding probabilities and the measures that are put in place by authorities in order to guarantee safe use of this alternative water source. One example of a good communication process is the production of the artisanal beer VIRA (Section 4.4) after reverse osmosis and the ultrafiltration of treated wastewater, as a proof-of-concept (not for commercial purposes but for visitors and events) by Águas do Tejo Atlântico utility, in Lisbon.

Notwithstanding its complexity, barriers and financial cost, water reuse appears to be a solution with a relevant potential to reduce urban ecological footprints. To promote implementation, it is pivotal to have more precise regulations regarding the quality requirement per type of reuse and economic and financial models that encourage and facilitate the practice.

In Portugal, the Strategic Plan for Water Supply and Sanitation for 2020, PENSAAR 2020 [55] negatively reviewed the previous goals established for urban wastewater reuse, mentioning in follow-up reports that the failure was due to "difficulties derived from the legal and regulatory frameworks of this activity, and Water Resource Tax levels that do not provide the necessary incentives for reuse, as well as subjective factors such as user lack of trust". As a comparison, Spain, with a comprehensive legislative framework (described in Section 3.4), has been able to implement water reuse projects with large volumes for agriculture, industry or urban irrigation, complementing other alternative sources, such as desalination.

Regarding the local dimension, local authorities and utilities should create a comprehensive and integrated water strategy that contains the following: (1) an analysis of climate-related risks and needs, (2) an analysis of the intensive-water-demand sectors and other urban strategies that are connected to water and wastewater management, as well as an evaluation of the impacts of a change in strategy (e.g., including water reuse); (3) a participatory analysis of local needs in terms of water resources; (4) a detailed explanation for all the components of the strategy and of the different elements with the identification,

description and quantification of the connected risks and benefits; and (5) transparent and effective communication of the identified solution, implementation schedule, risks and mitigations that have been identified.

## 6. Concluding Remarks

Water reuse provides a climate-independent water resource that could significantly impact the adaptation to climate change in the context of growing demand and increasing water scarcity. A comprehensive analysis of political, legal, economic and social acceptance factors must be performed to successfully implement water reuse projects, identifying gaps and drivers for success and devising strategies accordingly. Specifically, water reuse makes more sense when a large amount of consumption occurs near WWTPs and if there is steady demand over the year.

As water reuse's primary usage will be irrigation, the dimensioning of the systems will have to accommodate the higher demand in the drier, hotter season. Consequently, there will be a potential surplus in the wet season that can and should be used for environmental uses, such as aquifer recharge or the regulation of river and lake flows (both natural and artificial). Artificial lakes, particularly in urban scenarios, can also have a double function as retention basins for stormwater, helping to manage it and storing part of the water for the dry seasons ahead. In terms of environmental uses, aquifer recharge with reuse water should be considered a priority, as it contributes to balance supply and demand between dry and wet seasons, but also since this could improve groundwater systems in distress and under pressure from over-exploitation, particularly in urban settlements.

The presented advantages raise the importance of spatial planning and economic policies when addressing the location of water-intensive economic activities. If planned near WWTPs, it can help balance issues linked to water seasonality. On the other hand, when high-water-demand activities are clustered away from WWTPs, it is critical to create incentives for them to cooperate, develop synergies and co-finance the necessary water reuse infrastructure.

Finally, the legislative and governance steps are crucial for local authorities and utilities when trying to include water reuse in urban water management strategies. The case of the Portuguese legislative framework has seen recent improvements with the publication of a specific diploma on the water quality required for reuse. However, it lacks governance and business model instruments which could create positive economic incentives. In fact, although water reuse is often mentioned as desirable, its implementation is limited by specific reuse conditions that are not well defined, except for the conditions on quality. Therefore, to foster the adoption of policies that clearly define water reuse requirements, it would be relevant to legislate considering the state of the depletion of freshwater resources and its link with water reuse.

**Author Contributions:** S.C. and F.F. (Francesco Ferrario): Conceptualization, writing—original draft preparation; H.X.P. and F.F. (Filipa Ferreira): writing—review and editing; J.S.M.: supervision. All authors have read and agreed to the published version of the manuscript.

**Funding:** This research received no external funding.

**Institutional Review Board Statement:** Not applicable.

**Informed Consent Statement:** Not applicable.

**Data Availability Statement:** Not applicable.

**Acknowledgments:** The authors wish to thank Águas do Tejo Atlântico, S.A for providing information and images to illustrate the use cases presented in this paper.

**Conflicts of Interest:** The authors declare no conflict of interest.

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
