# Peer review of "Water Reuse, a Sustainable Alternative in the Context of Water Scarcity and Climate Change in the Lisbon Metropolitan Area"

_sustainability, doi:10.3390/su151612578_

Round 1

Reviewer 1 Report

The submitted article is more about policy related issues and close to general article. It doesn't have a research based  approach. All the figures are cited. But, from the prospective of general article, the article is well written and emphasized the use of wastewater in various aspects of human induced activities. 

However, there are few comments needed to be taken care of by the authors which are as follows:

1. What is single entry mentioned in line 189? Who will manage this and which technology will be used for such systems? It may be important for readers and may be incorporated in the manuscript.

2. Why 'tight compartmentalization' (line 203) will not work for wastewater reuse? It may be elaborated in the manuscript.

3. In urban areas, wastewater might be created not only by citizens, but also by industries. In some instances, the wastewater generated by industries  may be more hazardous. So, in line (227-233), how the extra cost on reusable water supply will be formulated. You can't pose extra cost evenly among all citizens. what may be the framework in such cases need to be discussed.

Author Response

Thank you for the revision of the manuscript and for your comments. We address them in the next paragraphs.

Regarding comment 1 (What is single entry mentioned in line 189? Who will manage this and which technology will be used for such systems? It may be important for readers and may be incorporated in the manuscript.)

This is a general characterization of ownership possibilities. A single entity managing water distribution and sewage treatment might be, from a business model and governance perspective, more favourable for project development. Who manages it and what technology is used is, for this argument, not relevant, as the point is that the governance model matters to the feasibility of projects and that the fact that there are several players with different strategies/policies involved, can hinder the development of water reuse projects. For this reason, legislative and regulatory incentives are needed to push the different actors towards the development of water reuse projects.

Regarding comment 2 (Why 'tight compartmentalization' (line 203) will not work for wastewater reuse? It may be elaborated in the manuscript.)

The conclusion in this sentence attempts to explain that: “tight compartmentalization of water supply and sanitation has the potential to result in poor institutional arrangements for global, integrated water cycle management in general, and water reuse in particular”, by which we mean (and the referred authors) that when many different entities are involved, their strategies and priorities might differ and therefore it’s more difficult to coordinate all the necessary parties in order to arrive at an integrated water management plan.

Regarding comment 3 (In urban areas, wastewater might be created not only by citizens, but also by industries. In some instances, the wastewater generated by industries may be more hazardous. So, in line (227-233), how the extra cost on reusable water supply will be formulated. You can't pose extra cost evenly among all citizens. what may be the framework in such cases need to be discussed.)

Highly polluting industries will have a WWTP integrated in their own facilities, mandatory under EU legislation. To avoid dispersion in this article, and for the sake of space, we did not explore in detail the possibilities when dealing with differing qualities of wastewater and their implication in a comprehensive cost recovery approach. The present article intends to present an overview of the main issues at hand and of the reality in the Lisbon Metropolitan Area and therefore, we do not attempt at proposing a specific taxing framework for water reuse, which could be an interesting follow-up for the article in fact. In the mentioned paragraph (227-233), we merely underline the need to have policy/legislation in place that helps keep water reuse costs at levels which incentivise project development and provide security for investments. This is addressed in the next paragraph (lines 234-241) and specifically mentions an industrial example, from the city of Prato, Italy, which we did not detail for the sake of space, but briefly mentioned in lines 255-260. It also highlights the need to reduce the gap between a direct cost-recovery approach and a more comprehensive one, which incorporates environmental and water security issues.

Reviewer 2 Report

GENERAL COMMENTS

Thank you for the opportunity to review this manuscript titled “Water Reuse, a sustainable alternative in the context of water scarcity and climate change in the Lisbon Metropolitan Area”.

The main aim of this paper is to explore the benefits and barriers of water reuse and intends to support local authorities in including water reuse in their water management strategies within a specific case study, in the Lisbon Metropolitan Area. Water scarcity is the topic of this study, and authors assess the possibility to introduce alternative water sources in all new project developments and existing ones. 

Results are well illustrated and explained. The presented work is the result of a comprehensive study that is relevant in the specific research sector. Along with the high quality of the work, I would like to highlight some minor comments and recommendations.

·       In the “Abstract”, and “Introduction” paragraphs it is recommended to underline the originality of the research results against the background of previously published decisions on the affected issues.

·       At the end of the "Introduction" section, it is recommended to improve a critical analysis of the approaches published in the works of other authors and, again, highlight the originality of the present work. 

Minor revisions.

Line 52: Please correct km3, the same typing error occurred in other lines or caption of the manuscript. 

Figure 1: Please translate the legend in English.

Author Response

Thank you for the revision of the manuscript and for you comments. Minor revisions you sent were also addressed in the revised manuscript.

We have made changes to the abstract and introduction to address your comments:

New abstract (new text proposal highlighted in yellow):

Water scarcity is a driver for society to rethink water management and change the paradigm of use with a fit-for-purpose approach - i.e., separating potable water for human consumption (drinking, cooking, personal hygiene) from all non-potable uses that do not require the same quality level. In this context, urban water reuse is a relevant tool for municipalities and metropolitan areas when dealing with pressure on water resources, among several alternative water sources that can be considered in a site-specific and integrated manner. Through the available literature and specific case studies in the Lisbon Metropolitan Area, this paper explores the benefits and barriers of water reuse and intends to support local authorities in including water reuse in their water management strategies. To the best of our knowledge, this is the first paper focusing on Portugal and the Lisbon Metropolitan Area, which globally examines governance, economic, legislative and social aspects regarding water reuse, and presents specific implementation examples covering potable and non-potable, direct and indirect reuse.

Addition of one last sentence in the Introduction (Section 1), after line 93:

There has been previous work on water quality standards or regional flows in the region, but the broader perspective we take on this article, encompassing governance, economic, legislative and social aspects, has not been sys-tematically approached, focusing on the Portuguese and Lisbon Metropolitan Area situation. As the effects of climate change, and the consequent water scarcity in Mediterranean climates, become harsher, there is a political will to create conditions that foster the use of alternative water sources. We aim to contribute to the discussion of the steps to be taken in order to make water reuse projects feasible for a more sustainable and integrated water management in the region.

In Line 52, the units were corrected to km3, and the legend of Figure 1 was also translated.

We hope this addresses your comments in a sactisfactory manner.

Thank you

Reviewer 3 Report

Comments and suggestions to authors

Authors:

 Sofia Cordeiro, Francesco Ferrario, Hugo Xambre Pereira, Filipa Ferreira  and José Saldanha Matos

Manuscript title:

 Water reuse, a sustainable alternative in the context of water  scarcity and climate change in the Lisbon Metropolitan Area

The topic presented in the article is very relevant to the need for environmental protection.

The decreasing amount of water with a growing economy is a global problem. Awareness of this threat should compel countries to reuse water primarily in agriculture, industry or urban irrigation. According to the principles of sustainable development, water management should play a key role in a circular economy situation.

The article presented here analyses the benefits and barriers of water reuse in the Lisbon metropolitan area in Portugal.

Author Response

Thank you for your revision and positive comment.